# Core Exercise as Non-Pharmacological Strategy for Improving Metabolic Health in Prediabetic Women

**DOI:** 10.3390/medicina61050942

**Published:** 2025-05-21

**Authors:** Nuray Yiğiter, Faruk Akçınar, Yavuz Yasul, Vedat Çınar, Taner Akbulut, Gian Mario Migliaccio

**Affiliations:** 1Department of Coaching Education, Faculty Sport Science, Inonu University, Malatya 44000, Türkiye; nuraydoogan@hotmail.com (N.Y.); faruk.akcinar@inonu.edu.tr (F.A.); 2Bafra Vocational School, Ondokuz Mayıs University, Samsun 55400, Türkiye; yavuz.yasul@omu.edu.tr; 3Department of Physical Education and Sport, Faculty Sport Science, Fırat University, Elazig 23000, Türkiye; cinarvedat@hotmail.com; 4Department of Coaching Education, Faculty Sport Science, Fırat University, Elazig 23000, Türkiye; takbulut@firat.edu.tr; 5Department of Human Sciences and Promotion of the Quality of Life, San Raffaele Rome Open University, 00100 Rome, Italy

**Keywords:** body composition, core training, glycemic control, insulin resistance, lipid profile, prediabetes

## Abstract

*Background and Objectives*: Prediabetes (PD) is characterized by impaired glucose metabolism and is associated with an elevated risk of type 2 diabetes and cardiovascular diseases. This study aimed to investigate the effects of an 8-week core exercise intervention on glycemic control, lipid profiles, insulin sensitivity, body composition, and physical performance in prediabetic women. *Materials and Methods*: Eighteen prediabetic women aged 20–55 years were randomly allocated to either a core exercise group (*n* = 9) or a control group (*n* = 9). The intervention group completed 24 supervised core exercise sessions over 8 weeks, whereas the control group remained sedentary. Pre- and post-intervention evaluations included anthropometric measurements, flexibility and strength tests, fasting and postprandial glucose levels, HbA1c, insulin, HOMA-IR, lipid profiles, and serum iron levels. Non-parametric tests were used for statistical analysis, and a Principal Component Analysis (PCA) and hierarchical clustering were conducted to explore multidimensional metabolic changes. *Results*: Core exercise significantly improved the body weight, BMI, fat percentage, and circumferences (shoulder, chest, and hip), along with an enhanced flexibility and back-leg strength (*p* < 0.05). Glycemic indices (FBG, PBG, and HbA1c), insulin, and HOMA-IR levels were significantly reduced, while serum iron and HDL-C increased (*p* < 0.05). Lipid markers, including the TG, LDL-C, CHOL, and TG/HDL-C ratio, showed significant improvements. The PCA and cluster analyses identified three clusters reflecting metabolic risk, body composition, and protective factors. *Conclusions*: This study demonstrates that an 8-week structured core exercise program significantly improves glycemic control, lipid profiles, insulin sensitivity, and body composition in women with prediabetes. Multivariate analyses (PCA and hierarchical clustering) corroborate a metabolic shift towards a reduced insulin resistance and a more favorable cardiometabolic profile, supporting core training as a viable, evidence-based non-pharmacological intervention to mitigate metabolic risk.

## 1. Introduction

Prediabetes (PD) is an intermediate stage in the progression of glycemic dysregulation and beta-cell dysfunction and serves as a critical precursor to diabetes mellitus (DM) [1]. PD is typically defined by a fasting plasma glucose (FBG) level of 100–125 mg/dL or an HbA1c value between 5.7% and 6.4%. Individuals with PD demonstrate elevated blood glucose levels that exceed normal ranges yet do not meet the diagnostic thresholds for DM, thereby placing them at an increased risk for metabolic deterioration [2]. The increasing prevalence of diabetes is largely driven by modern lifestyle factors, including the excessive consumption of processed foods, poor dietary habits, and physical inactivity. Consequently, the global diabetic population has surged from 150 million in 2000 to 450 million in 2019, with projections estimating a rise to 783 million by 2045, representing approximately one-eighth of the world’s population [3,4]. Beyond its classification as an endocrine disorder, DM is a major contributor to cardiovascular morbidity and mortality. Persistent hyperglycemia in both PD and DM patients accelerates adverse lipid profile alterations, increasing the susceptibility to coronary artery disease and atherosclerosis [5]. Without effective interventions, the burden of diabetes-related complications is expected to escalate significantly. In this context, structured physical activity is widely recognized as a non-pharmacological strategy for enhancing metabolic function, improving insulin sensitivity, and promoting glycemic control [6]. Previous studies have reported that many biochemical changes occur with exercise [7,8,9,10]. For example, different exercise modalities, including aerobic and resistance training, have demonstrated efficacy in reducing low-density lipoprotein cholesterol (LDL-C) and total cholesterol levels [11]. Additionally, high-intensity interval training (HIIT) has been associated with significant improvements in FBG [12] and reductions in hemoglobin A1c (HbA1c) levels [13]. Additionally, strength exercises prescribed for DM patients have shown positive outcomes in improving both FBG and cholesterol levels [14]. Although various exercise modalities have been extensively studied in metabolic regulation, the impact of core exercises on glycemic control and lipid metabolism in prediabetic individuals remains largely unexplored. As core training engages multiple muscle groups, enhances neuromuscular coordination, and improves metabolic efficiency, it may serve as an effective intervention for diabetes prevention. We hypothesized that an 8-week core exercise program will significantly improve glycemic regulation and lipid profiles in prediabetic individuals, leading to reductions in fasting blood glucose, HbA1c, insulin resistance, and adverse lipid markers, while enhancing metabolic and cardiovascular health. For this reason, this study aimed to determine the effects of core exercises, which are not a pharmacological agent, on the anthropometric, physical, and physiological indicators of individuals with prediabetes.

## 2. Materials and Methods

### 2.1. Participants

The current research was conducted with a pre-test–post-test C group experimental research design. Ethical approval was obtained from the Inönü University Ethics Committee on 8 May 2024 (Decision No: 2024-KAEK-08), and the study was conducted in accordance with the Declaration of Helsinki. Sample size determination using G*Power analysis indicated that a minimum of 8 participants per group was required to detect a statistically significant difference, assuming an alpha level of 0.05, statistical power (1 − β) of 0.80, an effect size of 1.79, and a two-tailed alternative hypothesis. Participants were selected based on FBG levels consistently ranging between 100 and 125 mg/dL over a 3-month period, in accordance with the diagnostic criteria for prediabetes established by the American Diabetes Association (ADA).

The exclusion criteria included individuals with cardiovascular or neurological disorders, pregnant women or those planning pregnancy, and individuals with orthopedic limitations preventing participation in exercise. Additionally, women using drugs that significantly affect carbohydrate metabolism (e.g., metformin, GLP-1 RAs, and SGLT-2 inhibitors) were also excluded. Consequently, 6 participants were excluded from the study before it began. During the study, an additional 6 participants withdrew due to relocation, bereavement-related absence, or a diabetes diagnosis requiring medication. These withdrawals were evenly distributed between the groups, with 3 participants leaving from the core exercise group and 3 from the C group, ensuring that group comparability was maintained.

The remaining participants (*n* = 18) were randomly assigned to the core exercise group (*n* = 9) or the C group (*n* = 9). Participants were aged between 20 and 55 years. The baseline assessments indicated that the mean height was 1.55 ± 0.52 m in the C group and 1.58 ± 0.41 m in the core exercise group. The mean body weight was 76.5 ± 15.00 kg in the C group and 74.8 ± 7.90 kg in the core exercise group. The BMI values were 32.3 ± 6.28 in the C group and 30.3 ± 4.28 in the core exercise group. All participants provided written informed consent prior to enrollment in the study.

This study was conducted using a post-test-only control group experimental design. Ethical approval was obtained from the Inönü University Ethics Committee on 8 May 2024 (Decision No: 2024-KAEK-08).

### 2.2. Experimental Design

All measurements were conducted in a controlled laboratory setting at two time points: baseline (pre-) and post-intervention. During both visits, participants provided a 6 cc intravenous blood sample at 9:00 a.m., following an 8–12 h overnight fast. To maintain consistency in timing, blood samples were collected at the same time point, ensuring that all samples were collected after the prescribed fasting period. Blood samples were collected in EDTA tubes, centrifuged at 4000 rpm for 10 min, and analyzed for FBG, HbA1c, serum iron, insulin, HOMA-IR, LDL-C, HDL-C, triglycerides (TG), total cholesterol (CHOL), and atherogenic index (TG/HDL-C). Additionally, demographic data, including age, height, and weight, as well as anthropometric measurements and physical performance metrics (flexibility and strength scores), were recorded. For postprandial blood glucose (PBG) measurements, a standardized meal was administered to all participants to ensure consistency in glucose response assessments. The meal consisted of 60% carbohydrates, 20% protein, and 20% fat, providing a total of approximately 500 kcal. The meal composition was designed to reflect a balanced macronutrient intake, minimizing variations in glycemic response due to differences in dietary composition. Postprandial blood glucose levels were measured on a different day at the same time point, exactly two hours after the ingestion of the standardized meal.

Participants in the core exercise group completed an 8-week structured core training program, consisting of three sessions per week (totaling 24 sessions). The exercise regimen included 10 distinct core exercises: plank, right/left side plank, reverse bridge, half push-ups, crunches, butterfly sit-ups, sit-ups, prone cobra, and half bridge. The program was designed based on the progressive overload principle and incorporates principle of progressive overload, incorporating incremental increases in duration, sets, repetitions, and rest intervals. Initially, the participants performed two repetitions of each exercise for 10 s in two sets, with progressive increases in subsequent weeks (Figure 1).

The C group did not engage in physical activity but attended the laboratory sessions as observers during the exercise interventions.

### 2.3. Anthropometric Measurements

*Body Mass*: body weight was measured using an InBody 230 electronic scale (±0.1 kg accuracy) with participants wearing light clothing and no shoes.

*BMI, Body Fat, and Muscle Mass*: These parameters were assessed using a bioelectrical impedance analyzer (InBody 120) under controlled conditions. Measurements were conducted in the morning (09:00–11:00) after a 12 h fasting period, with participants barefoot, free of metal objects, and post-restroom use. Body mass and body composition measurements were performed using separate devices to ensure precision and consistency. The height was recorded using a stadiometer [15].

*Circumference Measurements*: Measurements were obtained using a Gullick tape, ensuring no tissue compression and a relaxed muscular state. Shoulder: measured with arms extended laterally using a non-elastic tape. *Chest*: measured 2.5 cm above the nipple line, with arms relaxed after exhalation. *Waist*: measured at the umbilical level, with the abdomen relaxed. *Hip*: measured at the widest hip point near the symphysis pubis, with the participant upright and arms at their sides [16,17].

### 2.4. Performance Test

*Hand Grip Strength*: Assessed using a digital dynamometer (CAMRY EH101; Hengqi, China) following the American Society of Hand Therapists’ standard protocol. Participants sat with their elbows at 90 degrees, squeezing the dynamometer with maximum force for at least 2 s. After one practice trial per hand, three alternate trials were performed, starting with the dominant hand. The highest value was recorded [18]. *Back-Leg Strength*: Measured using a Takei Back-D dynamometer. Participants maintained straight knees and a flat back, pulling the dynamometer bar upward with maximum effort. Two trials were performed, and the highest value (kg) was recorded [19]. *Flexibility*: Evaluated using a sit-and-reach test bench. Participants sat barefoot with their legs extended, reaching forward without bending their knees. The furthest reach was held for 1–2 s, with two trials performed, and the highest value (cm) recorded [20].

### 2.5. Biochemical Analyses

HbA1c levels were determined according to the International Federation of Clinical Chemistry and Laboratory Medicine (IFCC) reference method. CHO, TG, and HDL-C levels were measured using enzymatic colorimetric methods, whereas LDL-C levels were calculated using the Friedewald equation: LDL-C (mmol/L) = CHO − HDL-C − (0.45 × TG). The atherogenic index was calculated as the TG/HDL-C ratio [21,22,23]. Insulin levels were measured using the electrochemiluminescence immunoassay (ECLIA) method on a Roche Cobas E 411 analyzer (Roche Diagnostics, Mannheim, Germany) [24]. FBG levels were determined using a glucose oxidase enzymatic analysis system (Roche Diagnostics). The estimated average glucose (eAG) was calculated using the following equation as previously described [23]: eAG (mg/dL) = (28.7 × HbA1c) − 46.7. Insulin resistance (HOMA-IR) was estimated using the Homeostatic Model Assessment for Insulin Resistance (HOMA-IR) formula by Park et al. [25]: HOMA-IR = [fasting insulin (μIU/mL)] × [fasting glucose (mmol/L)]/22.5. Serum iron levels were quantified using a spectrophotometric method, employing a reagent mixture composed of 200 mM citric acid, 34 mM ascorbic acid, 100 mM thiourea, and >3 mM ferene, prepared in a final volume ratio of 5:1:1 [26].

### 2.6. Statistical Analysis

All data were analyzed using a pre-test–post-test control group experimental design. The assumption of normality was assessed using the Shapiro–Wilk test. When the normality assumption was violated, non-parametric statistical methods were applied. Between-group differences in pre-test and post-test scores were evaluated using the Mann–Whitney U test, while within-group changes over time were analyzed using the Wilcoxon signed-rank test. All statistical analyses were performed using IBM SPSS Statistics 25.0, and the level of statistical significance was set at *p* < 0.05. Additionally, the relationships between variables and underlying data structures were explored using Principal Component Analysis (PCA) with biplot visualization, conducted via XLSTAT [27].

## 3. Results

According to Figure 2, the core exercise significantly impacted the body composition and anthropometric parameters compared with the C group, as observed through pre-test and post-test measurements. The core exercise group exhibited significant reductions in their body mass (*p* < 0.001), BMI (*p* < 0.001), shoulder (*p* = 0.022), chest (*p* = 0.041), and hip circumference (*p* = 0.007), along with a notable decrease in their body fat percentage (−14.49%, *p* = 0.005). However, the waist circumference did not significantly change. Conversely, the body muscle mass increased significantly in the core exercise group (*p* = 0.018), suggesting that the intervention effectively reduced fat mass while promoting muscle retention. The C group showed no significant changes across most parameters. These findings indicate that core exercise contributes to weight and fat loss while maintaining muscle mass, leading to a more balanced body composition.

According to Figure 3, core exercise significantly improved back-leg strength and flexibility while having no significant effect on hand grip strength. There were no notable changes in the right-hand and left-hand grip strength between the C or core exercise groups. However, the core exercise group showed a significant increase in back-leg strength (*p* = 0.015) compared with pre-test measurements, with a statistically significant difference between groups (*p* = 0.022). Similarly, flexibility scores improved significantly in the core exercise group (*p* = 0.037), with a notable difference between pre- and post-test results (*p* = 0.029). The C group did not exhibit any significant changes in any parameter. These findings suggest that core exercise effectively enhances lower body strength and flexibility, whereas its impact on hand grip strength remains minimal.

According to Figure 4, core exercise significantly improves metabolic health by reducing blood glucose levels and enhancing insulin sensitivity compared with the C group. The core exercise group exhibited a significant decrease in FBG (*p* < 0.001), PBG (*p* = 0.001), and HbA1c levels (*p* < 0.001), indicating better glycemic control. Additionally, serum iron levels showed a substantial increase (*p* < 0.001), suggesting improved iron metabolism. Insulin levels decreased significantly (*p* = 0.018), along with a marked reduction in HOMA-IR (*p* = 0.007), reflecting improved insulin resistance. In contrast, the C group did not exhibit significant changes in these parameters. These findings highlight the beneficial effects of core exercise on glucose metabolism, insulin regulation, and iron homeostasis, supporting its role in improving overall metabolic function.

According to Figure 5, core exercise significantly improved the lipid profile and reduced cardiovascular risk factors compared with the C group. The core exercise group showed a significant decrease in LDL-C levels (*p* = 0.047) and TG (*p* = 0.013), whereas HDL-C levels increased notably (*p* = 0.003), indicating a beneficial shift in lipid metabolism. The CHOL levels also decreased significantly in the core exercise group (*p* = 0.009), further supporting its positive effect on lipid regulation. Additionally, the atherogenic index was significantly reduced (*p* = 0.002), suggesting a lower risk of cardiovascular disease. In contrast, the C group did not exhibit significant changes in these parameters. These findings highlight the potential of core exercise for improving lipid metabolism and reducing cardiovascular risk.

The hierarchical clustering analysis using the unweighted pair group method with averages (UPGMA) identified distinct groupings among metabolic and anthropometric variables, categorizing them into three primary clusters: metabolic risk factors (Cluster A), body composition parameters (Cluster B), and cardiometabolic protective factors (Cluster C). Cluster A (blue group) comprised parameters linked to insulin resistance and cardiovascular risk, including CHOL, LDL-C, AIP, TG, HOMA-IR, insulin, PBG, HbA1c, and FBG. The co-occurrence of these variables within the same cluster suggests a shared pathophysiological mechanism underlying metabolic dysfunction, characterized by insulin resistance and an unfavorable lipid profile. The inclusion of the chest circumference in this cluster further highlights the association between visceral adiposity and metabolic syndrome. In this context, the presence of LDL-C in Cluster A reinforces its role as a key marker of cardiometabolic risk and its relevance to the pathogenesis of metabolic syndrome. Cluster B (orange group) comprised body composition and anthropometric indicators, including the BMI, body mass, hip circumference, body fat mass, shoulder width, waist circumference, and body muscle mass. The clustering of these variables highlights the strong interrelation between body composition and metabolic health. The association of the waist circumference with the body fat percentage and BMI particularly emphasizes the pivotal role of abdominal obesity in the pathogenesis of prediabetes. Cluster C (brown group) consisted of cardiometabolic protective factors, specifically HDL-C and iron levels. The inclusion of HDL-C reinforces its anti-atherogenic properties and its responsiveness to core exercise interventions. Additionally, the presence of iron, a critical component in oxygen transport, mitochondrial respiration, and muscle function, suggests its role in enhancing metabolic efficiency following exercise. Furthermore, iron has exhibited an inverse correlation with HbA1c levels and may influence the respiratory capacity via its role in hemoglobin synthesis and oxidative metabolism, thereby contributing to improved glycemic control and systemic oxygen utilization. These findings indicate that core exercise interventions facilitate a metabolic shift away from insulin resistance and dyslipidemia, while concurrently improving body composition and augmenting cardiometabolic protective factors. The results further suggest that exercise interventions may enhance metabolic health in prediabetic individuals by improving insulin sensitivity and glycemic regulation and reducing cardiovascular risk (Figure 6).

The Principal Component Analysis (PCA) identified a distinct separation between pre- and post-test conditions, indicating significant metabolic adaptations following the core exercise intervention. The first two principal components accounted for 64.93% of the total variance, whereas the first three components, with eigenvalues ≥ 1, explained 76.86% of the variance. The most substantial contributors to F1 were HOMA-IR (13.5%), HbA1c (12.6%), and TG (12.1%), followed by FBG, iron, PBG, insulin, HDL-C, AIP, CHOL, and LDL-C (10.1%, 9.2%, 9.0%, 8.9%, 7.1%, 6.7%, 6.4%, and 4.0%, respectively). F2 was predominantly influenced by AIP (23.4%) and HDL-C (16.3%), followed by insulin, CHOL, LDL-C, HOMA-IR, PBG, iron, FBG, and HbA1c (9.4%, 8.9%, 8.7%, 7.1%, 5.9%, 5.6%, 4.7%, and 2.4%, respectively). Participants positioned near the origin (0,0) demonstrated a higher degree of adaptation to the metabolic alterations induced by the core exercise intervention. In the pre-test phase, participants clustered within the orange region exhibited metabolic dysregulation, characterized by elevated HbA1c, FBG, HOMA-IR, insulin, CHOL, LDL-C, and AIP levels. These variables indicated an increased insulin resistance, impaired glucose homeostasis, and heightened cardiovascular risk associated with prediabetic profiles. Following the intervention, the core exercise group transitioned toward the blue region, distancing from these metabolic risk factors and aligning more closely with HDL-C and iron levels. This shift suggests a favorable metabolic response, marked by an improved lipid metabolism, enhanced insulin sensitivity, and increased oxygen transport capacity. The inverse relationship between the HDL-C and iron vectors relative to insulin resistance markers further implies a reduction in metabolic stress post-exercise (Figure 7).

## 4. Discussion

This study assessed the effects of core exercises on diabetes markers in patients with prediabetes (PD) and found that core exercise interventions resulted in significant changes across various parameters. The anthropometric findings are as follows: Compared with the C group, the core exercise demonstrated significant reductions in body weight, the BMI, and circumferences of the shoulder, chest, and hips in both intra-group and inter-group comparisons. However, despite the observed reductions in other body circumferences, no statistically significant change was detected in the waist circumference. We attribute this to the static and isometric nature of the core exercises, which may induce muscular tightening but might not be sufficient to yield substantial reductions in abdominal fat in an 8-week timeframe. Considering regional fat mobilization differences, especially the resistance of abdominal fat to reduction, a longer intervention may be necessary to observe significant changes. We acknowledge this as a limitation (Figure 2). The strength and flexibility findings are as follows: patients with PD who performed core exercises showed significant improvements in back-leg strength and flexibility, whereas no statistically significant changes were observed in these parameters in the C group (Figure 3). The diabetes markers and lipid profile findings are as follows: The core group exhibited significant reductions in plasma FBG, PBG, HbA1c, insulin levels, and the HOMA-IR index, alongside notable increases in iron and HDL-C levels. Between-group comparisons of post-test results reinforced these findings: the core exercise group exhibited significant decreases in their TG, total cholesterol, and TG/HDL-C atherogenic index, with a concurrent increase in HDL-C levels (Figure 4 and Figure 5). The hierarchical clustering analysis identified three main clusters: metabolic risk factors, body composition measures, and cardiometabolic protective factors (Figure 6). The PCA revealed significant metabolic shifts among the core exercise group participants, indicating a transition from a metabolically dysregulated state towards improved insulin sensitivity, lipid regulation, and oxygen transport. Notably, HDL-C was inversely associated with AIP, CHOL, TG, and LDL-C, emphasizing its distinct cardioprotective profile, whereas iron levels demonstrated a divergence from insulin-related markers, such as HbA1c, HOMA-IR, and insulin, suggesting a potential role in modulating insulin sensitivity and oxidative metabolism. These patterns, discernible through the vector direction and group clustering, underscore the multifactorial benefits of the core exercise intervention in prediabetic individuals (Figure 7). Orhan [28] emphasized the role of physical activity in influencing body composition, particularly the balance between muscle and fat mass, which critically affects the body fat percentage. König et al. [29] reported that 12 months of cycling ergometer exercises significantly reduced the body weight and BMI in individuals with prediabetes, attributing these changes to a decreased fat mass and increased muscle mass. Zhao et al. [30] observed notable weight and BMI reductions following combined exercise interventions in patients with type 2 diabetes (T2DM). Conversely, Gaitán et al. [31] found no significant changes in body weight, waist circumference, or fat mass after two weeks of cycling exercises in prediabetic individuals, linking the lack of effect to the short intervention duration and the influence of menopause on outcomes.

Bulguroğlu et al. [32] reported statistically significant improvements in the body weight, BMI, and waist-to-hip ratio after eight weeks of Pilates exercises in individuals with T2DM. Similarly, Fex et al. [33] demonstrated significant reductions in waist and hip circumferences in prediabetic and T2DM patients after 12 weeks of high-intensity interval training (HIIT). Kaplan [34] observed meaningful improvements in the body weight, BMI, fat percentage, and circumferences of the shoulders, chest, and hips in middle-aged women after 16 weeks of step-aerobic exercises. Furthermore, Rogers and Gibson [35] highlighted significant changes in body composition, particularly in chest and shoulder measurements, after an 8-week mat Pilates program. Collectively, these findings suggest that regular exercise positively impacts body composition and metabolic risk factors in individuals with prediabetes or diabetes. In particular, core exercises are emerging as an effective intervention for both individual and clinical applications. As previously noted, this study found that core exercises significantly improved back-leg strength and flexibility in patients with PD, although no changes were observed in the right- or left-hand grip strength in either group. Al-Sheerf et al. [36] reported significant increases in grip strength after six months of thrice-weekly, 40 min aerobic and resistance exercises in non-insulin-dependent diabetes patients. Sekendiz et al. [37] observed an increased flexibility and back strength in sedentary women following a 12-week core exercise program using a Swiss ball, while Cosio et al. [38] reported similar back strength improvements after a 5-week program. Hsieh et al. [39] found notable gains in leg strength in patients with T2DM undergoing 12 weeks of thrice-weekly resistance training at 75% 1-RM. Likewise, Aka et al. [40] demonstrated significant enhancements in back-leg strength and flexibility among sedentary women participating in reformer Pilates exercises. Hwang et al. [41] reported statistically significant improvements in flexibility after six weeks of thrice-weekly core and stretching exercises. Kirwan et al. [42] evaluated an 8-week online supervised exercise and education program for older adults with T2DM and reported significant reductions in waist circumference and increases in flexibility for both male and female participants. These findings underscore the efficacy of diverse exercise protocols in improving physical parameters and highlight the potential of core exercises to enhance fundamental abilities, such as back-leg strength and flexibility, thereby improving the physical performance and quality of life in patients with PD. This study also found that core exercises influenced plasma FBG, PBG, HbA1c, insulin levels, and the HOMA-IR index while significantly increasing iron levels. Specifically, the core exercise group exhibited reductions in plasma PBG, HbA1c, insulin levels, and HOMA-IR, alongside an increase in iron levels. Gaitán et al. [31] reported a significant decrease in PBG levels following a 16-week submaximal Vo_2_Max cycling ergometer test in prediabetic obese individuals. Similarly, Zhang et al. [11] demonstrated notable reductions in FBG, HbA1c, and postprandial glucose levels through moderate-intensity aerobic and resistance exercises in a meta-analysis. Studies tracking iron level changes have highlighted the critical role of iron in energy metabolism and its regulation via hepcidin and insulin, indicating a bidirectional relationship between glucose and iron metabolism [43,44]. Huang et al. [45] reported significant reductions in FBG, PBG, insulin, and HOMA-IR following a six-month program combining elastic band exercises and Yijinjing in prediabetic individuals.

Rahimi et al. [46] reported that Pilates and TRX exercises significantly reduced insulin levels. Similarly, Baker et al. [47], in a randomized controlled trial, found that aerobic and resistance exercises decreased HOMA-IR levels, although the impact on HbA1c depended on the exercise frequency. Özdamar et al. [48] observed that Nordic walking significantly lowered fasting blood glucose, insulin, and HbA1c levels in prediabetic individuals. Meta-analyses in the literature support that regular aerobic and resistance exercises improve HbA1c, FBG, and PBG levels [49,50,51]. In this study, insulin resistance was assessed using the HOMA-IR model, which is a widely used method in metabolic research. While HOMA-IR provides a practical estimate, it is an indirect measure based on fasting glucose and insulin levels. More precise methods, such as the euglycemic hyperinsulinemic clamp, directly assess insulin sensitivity but require invasive procedures and specialized equipment. Despite its limitations, HOMA-IR remains a valuable tool for evaluating exercise-induced metabolic changes in large-scale studies. In the current study, participants in the core exercise group demonstrated significant reductions in plasma LDL-C, TG, CHOL, and the TG/HDL-C atherogenic index, alongside a significant increase in plasma HDL-C levels. This indicates the efficacy of core exercises in improving plasma lipid profiles. Hou et al. [14] identified significant improvements in total cholesterol levels among individuals with type 2 diabetes mellitus (T2DM) following balance exercises, although no significant changes were noted in triglyceride levels. Yasul et al. [52] reported significant improvements in lipid metabolism among the exercise intervention groups. Total cholesterol (TC) levels were notably lower in the exercise groups than in the control, indicating a beneficial effect of physical activity on lipid regulation. Similarly, the atherogenic index of plasma (AIP) was reduced in the exercise groups, further supporting the role of structured training in improving cardiovascular health. These findings suggest that both moderate- and high-intensity exercise contribute to favorable lipid profile modifications and reduced cardiovascular risk. Öner et al. [22] demonstrated that a combination of core and HIIT exercises significantly reduced triglyceride levels but had no notable effect on LDL-C, HDL-C, or total cholesterol.

Çiçek et al. [53] noted significant reductions in LDL-C and total cholesterol levels after 16 weeks of core exercises, although HDL-C levels remained unchanged. Conversely, De Nardi et al. [54] found no differences in lipid profiles between HIIT and moderate-intensity continuous training (MICT). However, Zhang’s [11] meta-analysis revealed that moderate-intensity aerobic and resistance exercises significantly reduced LDL-C and total cholesterol levels. Aggarwala et al. [55] reported that a four-week aerobic exercise intervention lowered triglycerides but caused no significant changes in HDL-C or LDL-C levels. Magalhães et al. [56] observed that HIIT and resistance training positively influenced LDL-C and total cholesterol levels in individuals with T2DM. Similarly, Cai et al. [57] reported that elastic band resistance training reduced the total cholesterol and triglycerides while significantly improving HDL-C levels. Furthermore, exercise has been shown to exert significant benefits on lipid profiles, particularly through the reduction in LDL-C and TG levels, as well as a decrease in the TG/HDL-C atherogenic index [54]. Although HIIT, aerobic, and resistance exercises contribute to lipid profile improvement, core exercises appear to elicit more distinct and pronounced effects. Given these benefits, core exercises may play a crucial role in cardiovascular health by optimizing lipid metabolism, making them a valuable adjunct in the management of metabolic conditions such as prediabetes and diabetes.

### Strengths and Limitations

This study has several strengths that enhance its scientific rigor. The randomized controlled design improves its internal validity, whereas the comprehensive assessment of metabolic health—including glucose metabolism, lipid profiles, and insulin sensitivity—provides a multidimensional perspective. Standardized measurement protocols and advanced statistical analyses (PCA and hierarchical clustering) strengthen data reliability. The structured core exercise program ensures reproducibility, and the inclusion of serum iron levels provides novel insights into oxygen transport capacity.

However, limitations should be acknowledged. The small sample size may have affect generalizability, and the participant attrition reduced the statistical power. This study focused solely on prediabetic individuals aged 20–55 years, limiting its applicability to broader populations. The lack of an active C group prevents comparisons with other exercise modalities, and the unmonitored exercise adherence introduces variability. Additionally, dietary intakes and habitual physical activity were not controlled, which may have directly influenced the metabolic parameters examined. The macronutrient composition and caloric intake can modulate glucose metabolism, lipid profiles, and insulin sensitivity, whereas variations in daily physical activity levels outside the intervention may have contributed to individual differences in metabolic responses. The absence of objective tracking for these variables limits the ability to fully isolate the effects of the core exercise intervention. Future studies should address these limitations by expanding sample sizes, incorporating objective adherence monitoring, and controlling for external factors, such as diet and daily physical activity, to improve the accuracy of metabolic outcome assessments. Despite these constraints, the findings highlight core exercise as a viable non-pharmacological intervention for prediabetes management and cardiometabolic risk reduction.

## 5. Conclusions

This study demonstrated that core exercise effectively reduced body fat, BMIs, LDL-C, TG, and HOMA-IR while increasing HDL-C, iron, and insulin sensitivity. The PCA and hierarchical clustering analyses confirmed a metabolic transition from insulin resistance and dyslipidemia towards improved glucose regulation, lipid balance, and oxygen transport. The alignment of HDL-C and iron levels and improved metabolic markers suggests their role in counteracting cardiometabolic risks. These results highlight core exercise as a targeted intervention for prediabetes management, with potential applications in reducing cardiovascular risk. Future studies should examine long-term adherence and the integration of core exercise with dietary interventions for optimized metabolic outcomes.

## Figures and Tables

**Figure 1 medicina-61-00942-f001:**
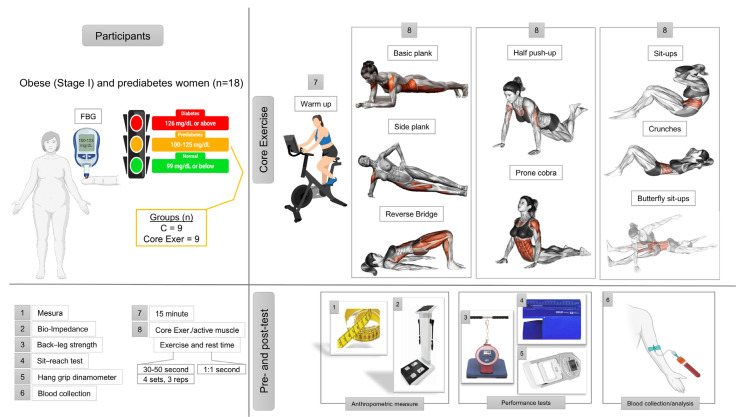
Illustrates the study design, including the participant characteristics, assessment protocols, and core exercise intervention.

**Figure 2 medicina-61-00942-f002:**
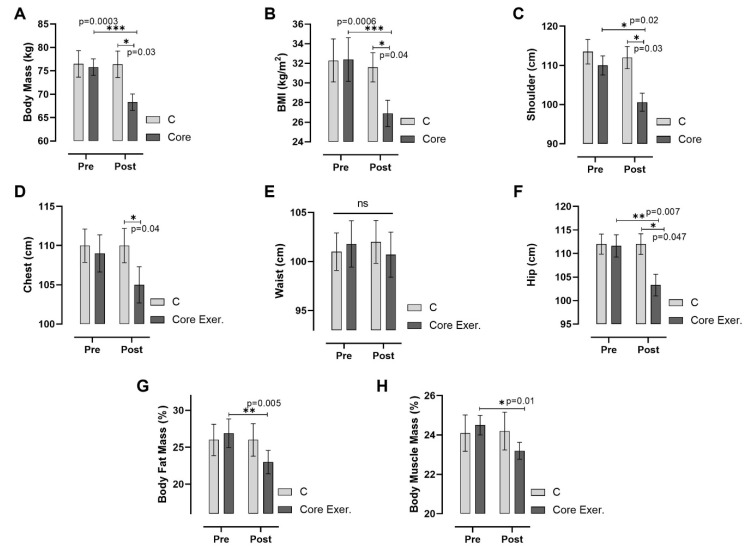
Changes in anthropometric parameters among individuals with prediabetes following the intervention are illustrated. Measurements include body mass (**A**), body mass index (BMI; (**B**)), shoulder circumference (**C**), chest circumference (**D**), waist circumference (**E**), hip circumference (**F**), body fat mass (**G**), and body muscle mass (**H**). The *x*-axis represents pre-test and post-test measurements, with post-test values statistically analyzed between the C and core exercise (Core Exer.) groups using an independent sample *t*-test. Horizontal lines above the bars indicate within-group differences over time. C: control group, Core Exer: core exercise group, ns: no significant difference, * *p* < 0.05, ** *p* < 0.01, and *** *p* < 0.001.

**Figure 3 medicina-61-00942-f003:**
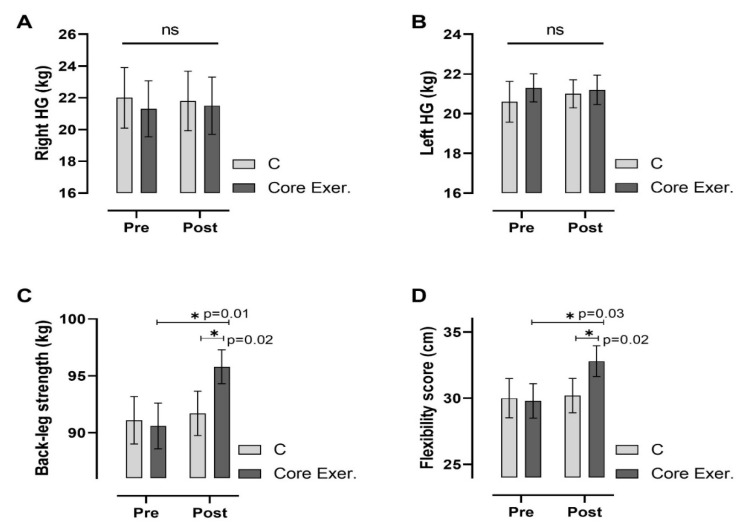
Changes in physical performance test outcomes among individuals with prediabetes following the intervention are illustrated. Measurements include right and left handgrip strength (**A**,**B**), back-leg strength (**C**) and flexibility score (**D**). The *x*-axis represents pre-test and post-test measurements, with post-test values compared between the C and Core Exer. groups using an independent sample *t*-test. Horizontal lines above the bars denote within-group differences observed between pre- and post-test values. C: control group, Core Exer: core exercise group, ns: no significant difference, and * *p* < 0.05.

**Figure 4 medicina-61-00942-f004:**
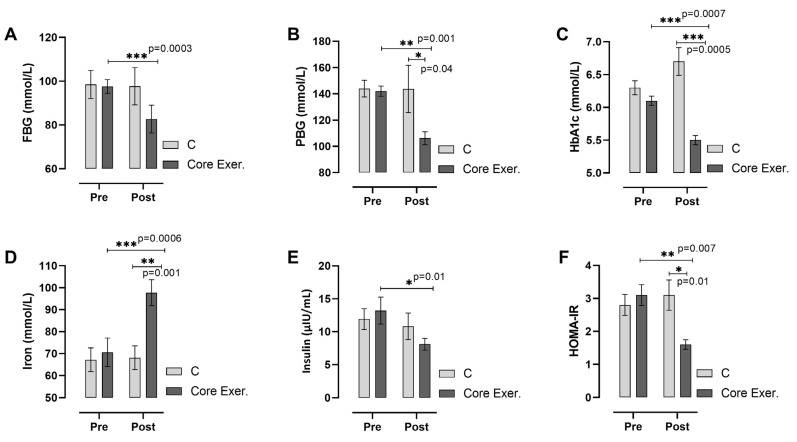
Changes in glycemic and insulin resistance markers among individuals with prediabetes following the intervention are illustrated. Measurements include fasting blood glucose (FBG; (**A**)), postprandial blood glucose (PBG; (**B**)), hemoglobin A1c (HbA1c; (**C**)), iron (**D**), insulin (**E**), and homeostatic model assessment of insulin resistance (HOMA-IR; (**F**)). The *x*-axis represents pre-test and post-test measurements, with post-test values compared between the C and Core Exer. groups using an independent sample *t*-test. Horizontal lines above the bars indicate significant within-group differences over time. C: control group, Core Exer: core exercise group, * *p* < 0.05, ** *p* < 0.01, and *** *p* < 0.001.

**Figure 5 medicina-61-00942-f005:**
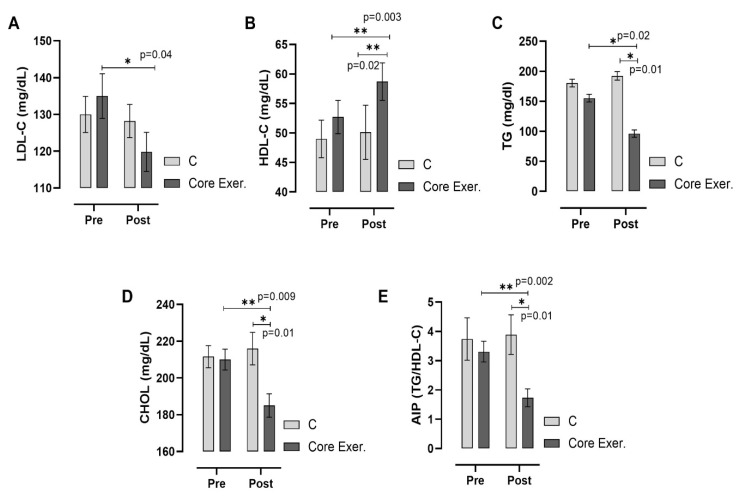
Changes in plasma lipid profiles among individuals with prediabetes following the intervention are illustrated. Measurements include low-density lipoprotein cholesterol (LDL-C; (**A**)), high-density lipoprotein cholesterol (HDL-C; (**B**)), triglycerides (TG; (**C**)), total cholesterol (CHOL; (**D**)), and atherogenic index of plasma (AIP; (**E**)). The *x*-axis represents pre-test and post-test measurements, with post-test values compared between the C and Core Exer. groups using an independent sample *t*-test. Horizontal lines above the bars highlight within-group variations from pre- to post-test. C: control group, Core Exer: core exercise group, * *p* < 0.05, and ** *p* < 0.01.

**Figure 6 medicina-61-00942-f006:**
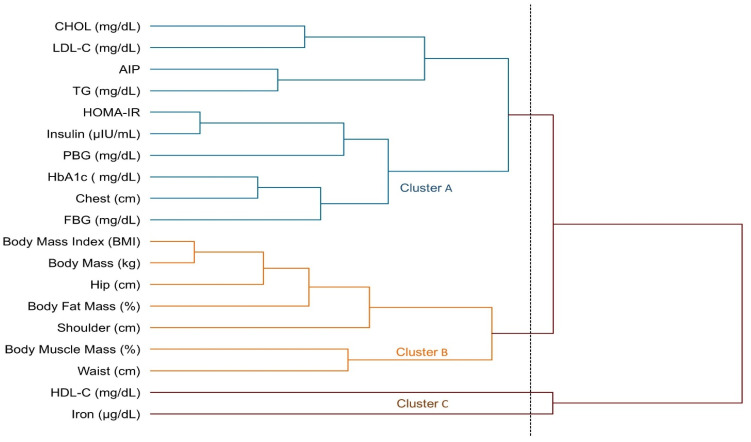
The hierarchical clustering of metabolic risk factors, cardiometabolic protective factors, and anthropometric parameters among individuals with prediabetes following the intervention is illustrated. Different colored lines indicate distinct clusters (A, B, and C). Cluster A includes metabolic risk factors, such as CHOL, LDL-C, AIP, TG, HOMA-IR, insulin, FBG, PBG, HbA1c, and chest circumference. Cluster B includes body composition parameters, including the BMI, body mass, hip circumference, body fat mass, shoulder width, body muscle mass, and waist circumference. Cluster C comprises cardiometabolic protective factors, specifically HDL-C and iron.

**Figure 7 medicina-61-00942-f007:**
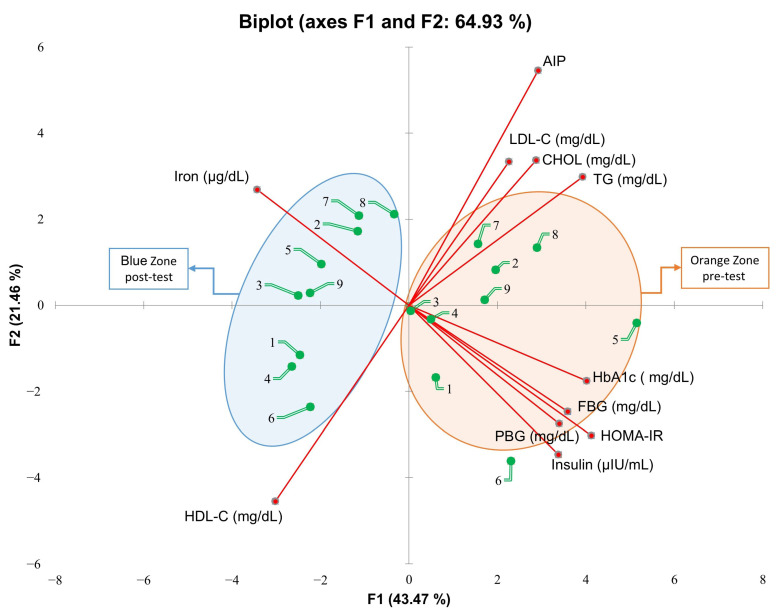
The Principal Component Analysis (PCA) biplot illustrating the metabolic and cardiometabolic changes among individuals with prediabetes following intervention. The orange zone (pre-test) clusters participants with elevated metabolic risk markers, including HbA1c, FBG, PBG, HOMA-IR, insulin, CHOL, LDL-C, TG, and AIP, indicating a profile associated with insulin resistance and dyslipidemia. Following the intervention, participants transition towards the blue zone (post-test), characterized by increased HDL-C and iron levels. Each participant’s position is represented by green-tailed spheres. Red vectors represent the contribution of each metabolic parameter to the principal components. The transition from the orange to the blue zone indicates a beneficial metabolic adaptation induced by core exercise. As only the core exercise group received the intervention, the PCA was restricted to this group, thereby ensuring analytical precision by excluding the untreated C group.

## Data Availability

The data presented in this study are available upon request from the corresponding author.

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
