# Peer review of "Core Exercise as Non-Pharmacological Strategy for Improving Metabolic Health in Prediabetic Women"

_medicina, 2025, doi:10.3390/medicina61050942_

Round 1

Reviewer 1 Report

Comments and Suggestions for Authors

All comments, suggestions, and questions are available throughout the manuscript.

Author Response

Reviewer 1

Line 41-42: Please authors, give numerical level that is considerate prediabete stage.

  • Response: Thank you for your valuable comment. We agree that providing specific numerical criteria for the prediabetic stage would improve the clarity and precision of the manuscript. We have added the relevant fasting glucose and HbA1c thresholds according to international guidelines (Lines 41-44; page 1).

Line 105-106 and 111-112: Please author, you can considere the first or second period.

  • Response: Thank you very much for your thoughtful comment. We appreciate your suggestion and have made an effort to clarify the methodology in the manuscript. To address your point, we have specified that measurements were conducted at both the baseline (pre) and post-intervention time points. Additionally, to ensure consistency in blood collection timing, all samples were drawn at 9:00 a.m. following an 8–12 hour overnight fast. We hope this revision helps to clarify the timing of the blood samples, and we are open to any further suggestions you may have (Lines 105-110; page 3).

Line 199: Please author, present the percentage value.

  • Response: Thank you very much for your helpful comment. In response to your suggestion, we have added the exact percentage change in body fat to provide a clearer understanding of the observed difference. We appreciate your attention to detail (Lines 199; page 6).

Line 251: Please authors, uniformize your data presentation if is p=0.01 or p=0.02, compare with HDL-C result (p=0.003 or p=0.02.

  • Response: Thank you very much for your valuable observation. In accordance with your suggestion, we have standardized the presentation of all p-values throughout the manuscript. Specifically, we now report all p-values with three decimal places (e.g., p=0.018, p=0.002, p<0.001) for consistency and clarity, including the result for HDL-C. We appreciate your careful review, which helped us improve the uniformity and precision of the data presentation (Lines 255; page 9).

Line: 273: Please author, what can you tell about the LDL-C in the cluster A?

  • Response: Thank you for your insightful comment. As observed, LDL-C is included in Cluster A, which encompasses parameters associated with insulin resistance and cardiovascular risk. We have revised the manuscript to clarify this point as follows:

Cluster A (blue group) comprises parameters linked to insulin resistance and cardiovascular risk, including CHOL, LDL-C, AIP, TG, HOMA-IR, insulin, PBG, HbA1c, and FBG. The co-occurrence of these variables within the same cluster suggests a shared pathophysiological mechanism underlying metabolic dysfunction, characterized by insulin resistance and an unfavorable lipid profile. The inclusion of chest circumference in this cluster further highlights the association between visceral adiposity and metabolic syndrome. In this context, the presence of LDL-C in Cluster A reinforces its role as a key marker of cardiometabolic risk and its relevance to the pathogenesis of metabolic syndrome (Lines 280-281; page 10).

Line 282: Please authors, iron is correlated with HbA1c, respiratory, consider it.

  • Response: We sincerely thank the reviewer for the insightful comment. We agree that iron plays a multifaceted physiological role beyond its traditional metabolic functions. We have expanded the description of Cluster C in the revised manuscript to highlight the inverse relationship between iron and HbA1c, as well as iron’s involvement in respiratory and oxidative metabolism through its role in hemoglobin synthesis and mitochondrial function. These modifications acknowledge iron’s emerging role in glycemic control and systemic oxygen utilization, particularly relevant in the context of exercise-induced metabolic adaptations (Lines 292-295; page 11).

Line 342: Please authors, explore the correlation between iron to group formed by insuline, and HDL-C to TG and LDL-C group.

  • Response: We appreciate the reviewer’s insightful suggestion and have revised both the figure legend and discussion section accordingly. In the PCA biplot (Figure 7), we now explicitly describe the inverse spatial orientation of HDL-C relative to TG and LDL-C, reinforcing HDL-C’s opposing behavior to atherogenic lipids. Furthermore, iron's vector projects in the opposite direction from insulin, HOMA-IR, and other glycemic markers, suggesting an inverse relationship consistent with its known involvement in mitochondrial function and insulin sensitivity. These vector relationships have been clarified in the revised figure legend, and the rationale for including only the core exercise group in PCA analysis due to the lack of intervention in the control group—has also been added to avoid potential confusion (Lines 331-332; page 12, Lines 357-363; page 13).

Reviewer 2 Report

Comments and Suggestions for Authors

Thank you for the opportunity to review the article entitled "Core Exercise as a Non-Pharmacological Strategy for Improving Metabolic Health in Prediabetic Individuals".

This paper investigates the impact of core exercises on the anthropometric, physical, and physiological metrics of persons with pre-diabetes.

The abstract and the Introduction section briefly but effectively introduce the reader to the issues discussed in the rest of the manuscript - I have no objections to these sections.

In the methodological section, I would like the authors to address the following comments:
1) Did the authors remove from the research patients with obesity who were taking drugs that significantly influence carbohydrate metabolism, yet were not utilized due to disturbances in this metabolism (such as metformin, aGLP-1, iSGLT-2)? If that is the case, it would be warranted to include such information.
2) Based on what guidelines were only patients with glycemia >140 mg/dl considered to be patients with impaired carbohydrate metabolism? In fact, normative glycemia values ​​are in the range of up to 99 mg/dl.
3) If all the study participants were female, it would be reasonable to change the title to reflect the fact that only women were studied.

In the results section:
1) In certain graphs within Figure 2, Figure 4, and Figure 5, there is an absence of dual evaluation of statistical significance regarding differences—was this deliberate or an oversight?
2) In Figure 4B, there is a typographical error - TBG should be replaced with PBG. This requires correction. Additionally, please examine the entire article for other minor typographical errors.
In addition to the aforementioned, the findings section does not provide any significant objections; it is meticulously conceived and articulated.

The discussion part appears to be well-conceived, and I have no significant objections; nonetheless, I find the authors' lack of exploration about the ineffectiveness of the proposed intervention on waist circumference to be a notable omission. I recommend that the authors address this issue. The objective assessment of the study's shortcomings and strengths warrants acknowledgment.

The conclusions are formulated correctly, I have no reservations about them.

The study generates curiosity and, albeit necessitating some revisions, possesses significant potential to engage the scientific community.

I will willingly conduct second review upon receiving a response from the authors.

Author Response

  1. Did the authors remove from the research patients with obesity who were taking drugs that significantly influence carbohydrate metabolism, yet were not utilized due to disturbances in this metabolism (such as metformin, aGLP-1, iSGLT-2)? If that is the case, it would be warranted to include such information.

Response: Thank you for your insightful comment. Yes, we excluded patients who were taking medications known to significantly influence carbohydrate metabolism, such as metformin, GLP-1 receptor agonists, and SGLT-2 inhibitors, even if they were not used for diabetes mellitus but for other metabolic indications. This exclusion was applied to avoid any confounding effect on the metabolic parameters under investigation. We have now clarified this point in the "Materials and Methods" section of the manuscript (Lines 88-90; page 2).

  1. Based on what guidelines were only patients with glycemia >140 mg/dl considered to be patients with impaired carbohydrate metabolism? In fact, normative glycemia values ​​are in the range of up to 99 mg/dl.

Response: We appreciate the reviewer’s observation and agree that, American Diabetes Association (ADA) guidelines, prediabetes is defined as a fasting blood glucose (FBG) level of 100–125 mg/dL. In the original draft, we erroneously stated participant inclusion based on an FBG of 140–199 mg/dL. We have now corrected this to align with ADA criteria, and participants included in this study had FBG values consistently within the 100–125 mg/dL range. The manuscript has been revised accordingly to reflect this correction (Lines 82-85; page 2).

  1. If all the study participants were female, it would be reasonable to change the title to reflect the fact that only women were studied.

Response: We thank the reviewer for this valuable suggestion. All participants included in the final sample of our study were indeed female. Accordingly, we have revised the manuscript title to accurately reflect the participant characteristics by indicating the gender of the participants. The updated title now reads:

"Core Exercise as a Non-Pharmacological Strategy for Improving Metabolic Health in Prediabetic Women"

This change has been implemented in the title section of the manuscript to ensure clarity and precision (Lines 3; page 1).

  1. In certain graphs within Figure 2, Figure 4, and Figure 5, there is an absence of dual evaluation of statistical significance regarding differences was this deliberate or an oversight?

Response: We sincerely thank the reviewer for this insightful comment. The absence of dual evaluation of statistical significance in certain graphs within Figures 2, 4, and 5 was indeed a deliberate decision rather than an oversight. Our primary objective in this regard was to preserve visual clarity and to emphasize the most salient, robust findings. Including exhaustive significance annotations across all possible comparisons in these figures would have risked visual overcrowding, potentially detracting from the interpretability of key results. Furthermore, we consciously reserved the comprehensive statistical comparisons and broader interpretive context for the PCA and hierarchical clustering analyses. These multivariate techniques were specifically selected to provide a holistic view of the dataset, allowing for the visualization of overarching patterns, intergroup relationships, and individual variability in a more integrated manner. By doing so, we aimed to ensure that readers could engage with the full scope of the analysis while maintaining focus in the more targeted figures. We hope this clarifies our rationale, and we remain grateful for the reviewer’s attention to detail and constructive feedback.

  1. In Figure 4B, there is a typographical error TBG should be replaced with PBG. This requires correction. Additionally, please examine the entire article for other minor typographical errors. In addition to the aforementioned, the findings section does not provide any significant objections; it is meticulously conceived and articulated.

Response: We sincerely appreciate your careful review. The typographical error in Figure 4B has been corrected “TBG” has been replaced with “PBG” as noted. In addition, the entire manuscript has been thoroughly reviewed for other minor typographical errors, and appropriate corrections have been made. Thank you once again for your valuable feedback (Figure 4B; page 8).

  1. The discussion part appears to be well-conceived, and I have no significant objections; nonetheless, I find the authors' lack of exploration about the ineffectiveness of the proposed intervention on waist circumference to be a notable omission. I recommend that the authors address this issue. The objective assessment of the study's shortcomings and strengths warrants acknowledgment.

Response: We appreciate the reviewer’s valuable observation regarding the lack of detailed discussion on the ineffectiveness of the intervention on waist circumference. As rightly pointed out, although our core exercise intervention resulted in some tightening due to the isometric static stretching structure of the exercises, it did not produce a statistically significant change in waist circumference.
We accept this limitation and acknowledge that, while other anthropometric measurements showed positive developments, the lack of significant change in waist circumference may stem from regional differences in fat reduction and the known difficulty of achieving localized fat loss in the abdominal area. We believe that a longer intervention period might produce different outcomes, given the trends observed in other measurements.
Admittedly, we hesitated to include a strong statement on this matter due to concerns over drawing overly definitive conclusions from a relatively short intervention. We have addressed this in the revised manuscript to provide a more balanced discussion of the study’s strengths and shortcomings (Lines 338-344; page 12).
